# Methylphenidate reorganizes cortical hierarchy through dopaminergic modulation

Dardo Tomasi [1] ✉, Peter Manza [1,2], Şükrü Barış Demiral [1], Weizheng Yan [1], Kylee B. Miller [1], Faith Veenker[1], Joshua Zhao[1], Christina Lildharrie[1], Michele-Vera Yonga[1], Sarah Abey[1], Michaelene VanDine[1], Gene-Jack Wang [1] & Nora D. Volkow [1]

Dopaminergic signaling shapes large-scale brain network architecture, constraining neural communication along a principal gradient that spans unimodal sensorimotor to transmodal association cortices. While more differentiated gradients are typically linked to enhanced cognition, it remains unclear whether dopamine-enhancing psychostimulants, such as methylphenidate (MP), amplify or compress this functional hierarchy to support attention. Across two double-blind, placebo-controlled studies in healthy adults (n = 38 and n = 20), we combined 60 mg oral MP with PET and fMRI to assess striatal dopamine function and cortical organization. MP consistently compressed the principal gradient, reducing segregation between sensory and association areas. The degree of compression predicted individual variation in striatal D1 and D2 receptor availability. MP-induced gradient compression in inferior parietal cortex tracked attention improvements. Critically, we validated key findings in a large, independent cohort from the Adolescent Brain Cognitive Development (ABCD) study (n = 4,958). These results highlight a dopamine-sensitive mechanism linking cortical functional reorganization with cognitive performance.

Dopamine (DA) signaling in the striatum is crucial for motor control, cognitive function, motivation, and reinforcement learning[1]. The striatum receives dense dopaminergic projections from midbrain nuclei[2], with DA exerting its effects via D1-like (D1R) and D2-like (D2R) receptors[3], which are expressed in distinct populations of medium spiny neurons[4,5] that indirectly project broadly to cortical and subcortical targets. Disruptions in dopaminergic signaling are implicated in neuropsychiatric disorders such as attention-deficit/hyperactivity disorder (ADHD), schizophrenia, and substance use disorders[6–8]. Understanding how pharmacological agents that modulate DA signaling alter brain-wide functional architecture is therefore critical for clarifying the neural mechanisms underlying these conditions.

Methylphenidate (MP), a first-line treatment for ADHD, blocks DA and norepinephrine (NE) transporters (DAT and NET) thereby increasing extracellular DA and NE in the striatum[9]. PET studies have shown that clinically relevant doses of MP given to healthy volunteers lead to 10–20% reductions in D2R availability, reflecting increases in extracellular DA[10]. DA and NE enhance performance on tasks requiring attention, working memory, and cognitive control[11], and modulate large-scale network dynamics, particularly in prefrontal regions[12]. However, the broader impact of MP's enhancement of DA on large-scale cortical organization and hierarchical functional architecture remains poorly understood.

[1]National Institute on Alcohol Abuse and Alcoholism, National Institutes of Health, Bethesda, MD, USA. [2]Kahlert Institute for Addiction Medicine, University of Maryland, Baltimore, MD, USA. ✉e-mail: dardo.tomasi@nih.gov

Emerging evidence suggests that neuromodulators such as DA shape the macroscale hierarchical organization of the cortex[13]. A principal gradient of functional connectivity (a dominant axis of brain organization spanning primary sensorimotor to transmodal association cortex) explains a substantial portion of variance in resting-state fMRI (rsfMRI) data (30–60%)[14]. This gradient reflects the topographical progression from externally oriented to internally oriented processing[15], and aligns with molecular gradients, including D1R expression, which increases along the hierarchy from sensory to associative regions[13]. We hypothesized that by enhancing DA signaling, MP would compress this gradient, reducing the functional segregation between unimodal and transmodal regions and shifting cortical integration toward association areas.

Although prior studies have shown that MP alters global functional connectivity in somatomotor, visual, and cognitive control networks[16], its effects on cortical gradients remain unexplored. Moreover, the roles of striatal DA receptor availability (D1R and D2R) and MP-induced DA increases in changes to the principal gradient are currently unknown.

DA's role in modulating cortical integration is well established[13,17]. MP increases striatal DA[18], and enhances functional segregation between the default mode (DMN) and task-positive networks[19] suggesting it may also alter hierarchical differentiation between cortical systems. Gradient compression could facilitate cross-network communication by reducing functional boundaries between specialized regions, potentially enhancing cognitive flexibility and integration. These effects are consistent with MP-induced increases in frontoparietal network connectivity[20].

The hierarchical architecture of the brain can be captured using rsfMRI and diffusion embedding, a nonlinear dimensionality reduction technique that identifies continuous spatial gradients in functional connectivity[21]. The principal gradient derived from this method is robust across datasets[22,23] and closely correlates with spatial patterns of gene expression, intracortical myelin, and cortical thickness[24], offering a powerful framework for studying the impact of dopaminergic modulation on cortical organization.

Here, we investigated how MP alters cortical hierarchy and its relationship to striatal DA signaling with PET and rsfMRI (42 min total scan time) for precision functional mapping[25] in healthy adults. We used diffusion embedding to derive functional gradients under placebo (PL) and MP and assessed reproducibility in an independent cohort of 20 participants. Using partial least squares (PLS) regression, we examined how MP-induced changes in cortical gradients relate to striatal DA receptor availability and DA increases, operationalized as reductions in [¹¹C]raclopride's specific binding. We hypothesized that MP would compress the principal gradient, reduce the segregation between unimodal and transmodal cortex, and that gradient reorganization would predict individual differences in striatal D1R and D2R. To test the developmental generalizability of our findings, we hypothesized that children with ADHD receiving stimulant medications in the ABCD study would show a similar compression of the principal cortical gradient.

## Results

The 38 healthy adults in the Discovery cohort underwent 3 PET scans on different days. One [¹¹C]NNC-112 scan and two [¹¹C]raclopride scans were used to measure dopamine D1R and D2R receptor availability, respectively (Fig. 1a and b). The [¹¹C]raclopride scans were conducted 1 h after ingesting a 60 mg oral dose of MP on one study day, and a placebo pill on the other study day. The order of the MP and PL [¹¹C]raclopride sessions was randomized in a counterbalanced fashion across participants. During the subsequent MRI session, three consecutive fMRI scans at 3 Tesla were used to assess brain functional connectivity (FC) at rest. Study days were scheduled one week apart. The participants were blind to the type of pill given on each study day. There were no significant differences in head motion between PL and MP, as assessed by framewise displacement (FD; PL: 0.118 ± 0.040 mm, MP: 0.113 ± 0.036 mm; $P = 0.3$, 2-sided paired t-test).

### High-dimensional connectivity gradients

Using the diffusion embedding method[15], we mapped FC gradients to capture the principal gradient of macroscale cortical organization

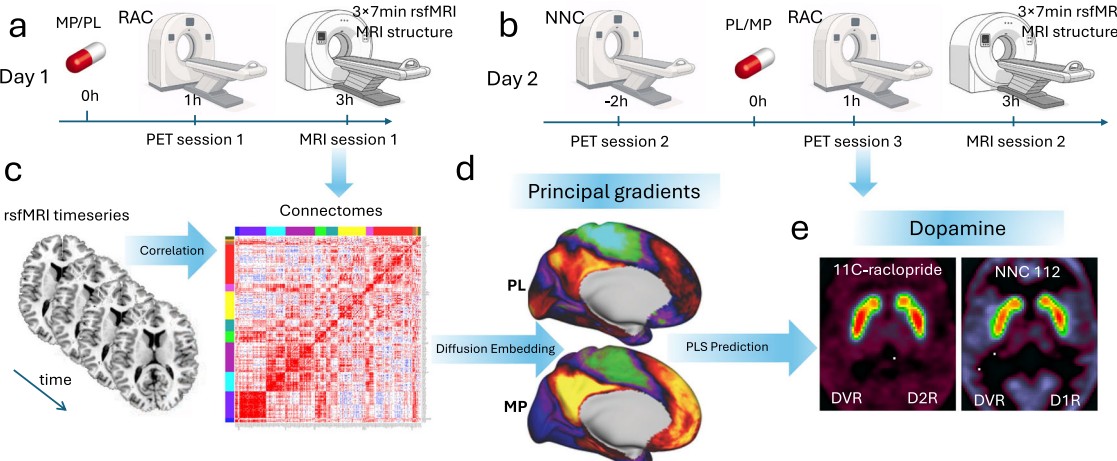

**Fig. 1 | Study design.** Thirty-eight healthy adults (Discovery cohort) participated in two study sessions on separate days. **a** On one study day, participants received an oral dose of either 60 mg methylphenidate (MP) or a placebo (PL), followed by a [¹¹C]raclopride PET scan to assess D2 receptor (D2R) availability. Afterward, they underwent three resting-state fMRI (rsfMRI) scans to evaluate functional connectivity (FC). **b** On the second study day, participants first underwent a [¹¹C]NNC-112 PET scan to quantify D1 receptor (D1R) availability. They then repeated the D2R PET and rsfMRI scans after receiving the alternate drug (PL or MP), with session order counterbalanced. **c** FC was estimated using Pearson correlation, generating average dense connectomes with 91,282 grayordinates for the MP and PL conditions, and parcellated connectomes with 438 parcels per each rsfMRI scan. Cosine similarity matrices were then constructed by retaining the top 10% of positive edges per row of the connectomes. **d** Diffusion embedding was applied to map the principal and secondary gradients of the brain's hierarchical organization. **e** Partial least squares (PLS) regression with leave-one-out cross-validation (LOO-CV) was used to predict PET-derived metrics (D1R, D2R, and MP-related dopamine increases) based on principal and secondary gradients across individuals.

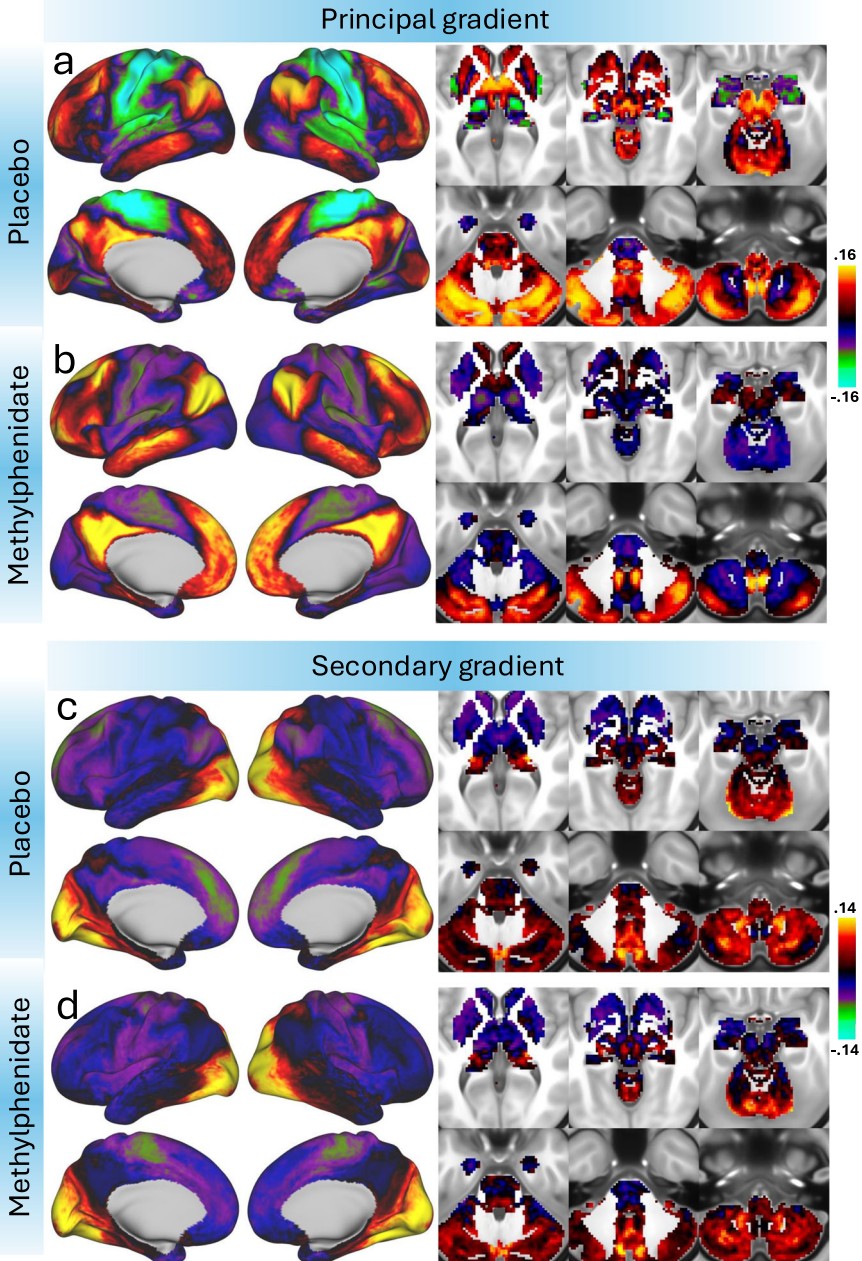

**Fig. 2 | High-dimensional gradients of cortical and subcortical organization under placebo and methylphenidate conditions.** Average principal (**a**, **b**) and secondary (**c**, **d**) gradients mapped onto the dorsal and medial cortical hemispheres (left) and six axial slices covering subcortical brain regions (right). These average gradients were computed separately for the placebo (**a**, **c**) and methylphenidate (**b**, **d**) conditions in 38 healthy adults. The principal gradient reflects a macroscale axis of cortical hierarchy, transitioning from sensorimotor to association regions, whereas the secondary gradient captures distinctions between visual to association regions. Gradients were derived for 91,282 grayordinates using diffusion embedding with the mapalign Python package and displayed with the connectome workbench viewer.

from rsfMRI timeseries. We first constructed average connectivity matrices, that reflect the temporal correlation among 91,282 grayordinates in the brain, independently for PL and MP (Fig. 1c). Retaining the top 10% of positive edges per row we constructed similarity matrices and using the cosine distance we computed functional gradients with the mapalign python package. Two gradients were then extracted for each drug condition (Fig. 1d).

For the PL condition, Fig. 2a shows that the principal gradient of cortical organization exhibited a characteristic hierarchical pattern, with the lowest values observed in unimodal sensory regions (e.g., somatosensory and primary visual cortices) and the highest values in transmodal association areas (e.g., DMN and the frontoparietal network, FPN). The gradient showed a smooth transition along the cortical axis, aligning with previously established macroscale functional hierarchies[15]. The second principal gradient captured a visual-to-association hierarchical axis (Fig. 2c), spanning from occipital visual regions to prefrontal and somatomotor cortices.

In subcortical structures and under PL, the principal gradient reflected a hierarchical axis spanning from sensorimotor-associated regions to higher-order associative areas (Fig. 2a). This gradient spanned from the sensorimotor territories of the striatum (posterior putamen) and thalamus (medial ventroposterior and pulvinar regions) to associative territories of the caudate, and pallidum. In the thalamus, the gradient followed a dorsal-to-ventral and posterior-to-anterior

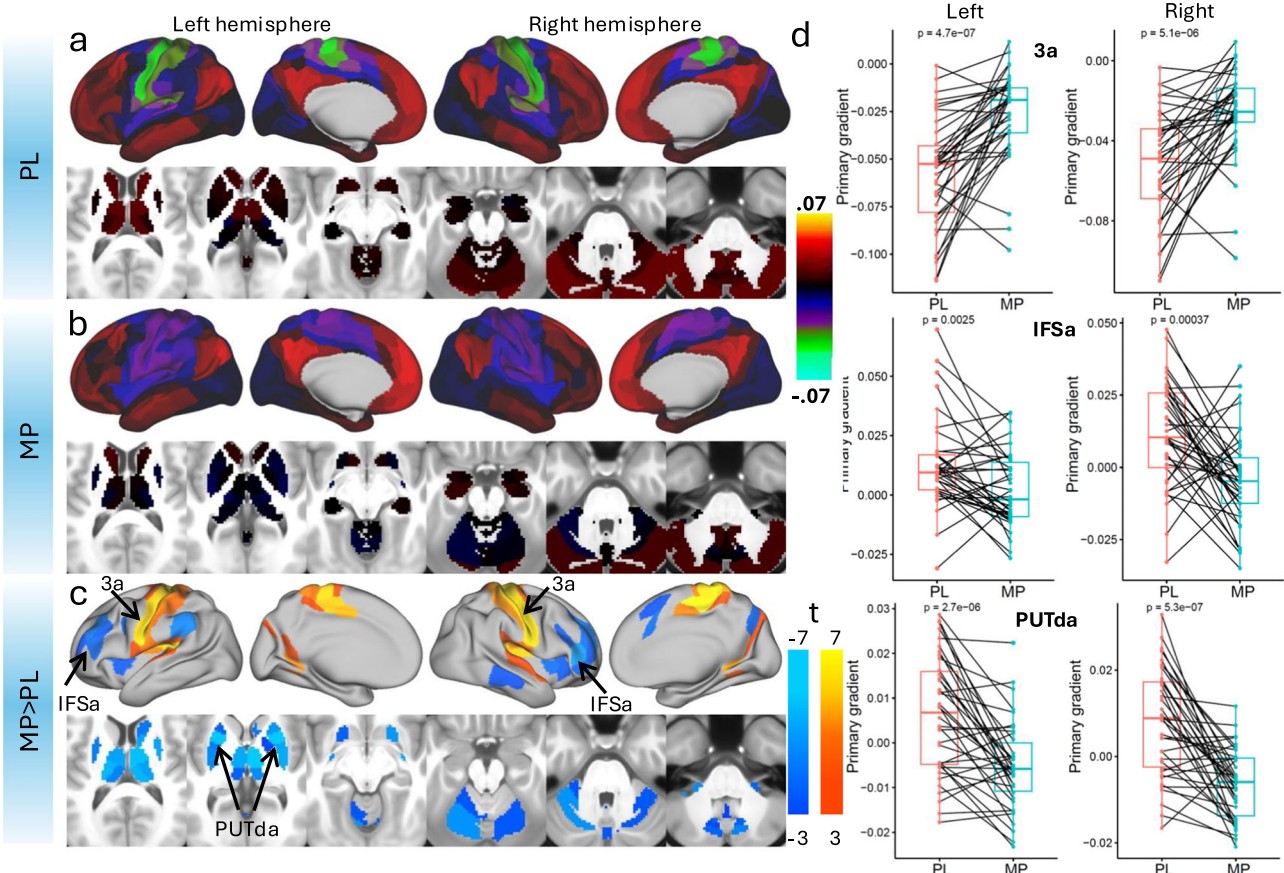

**Fig. 3 | Effect of MP on the principal gradient of brain functional organization.** Surface renderings display the strength of the normalized principal gradient overlaid on lateral and medial views of the left and right hemispheres, along with six axial slices covering subcortical regions, for the placebo (PL; **a**) and methylphenidate (MP; **b**) conditions in 38 healthy adults. The parcellation atlas included 438 cortical and subcortical regions. **c** Statistical difference maps (t-scores) illustrate regions with significant changes in gradient strength between PL and MP, estimated using a linear mixed-effects (LME) model with age, sex, race, body mass index, and intelligence as covariates. The statistical maps are displayed using a false discovery rate (FDR) threshold of $p < 0.05$. **d** Paired plots show individual differences in principal gradient values under PL and MP in three bilateral regions of interest: the primary somatomotor cortex (3a), the inferior frontal sulcus (IFSa), and the dorsal anterior putamen (PUTda); each data point represents one biological replicate ($n = 38$ healthy adult participants), with matched values under both PL and MP conditions. Each box shows the median (center line), the interquartile range (IQR; bounds of the box representing the 25th and 75th percentiles), and the minimum and maximum values within 1.5 times the IQR (whiskers). All paired t-tests are two-sided. Source data are provided as a Source Data file.

organization. The principal gradient also exhibited low values in the amygdala and hippocampus, positioning these regions closer to sensorimotor-associated subcortical structures rather than higher-order associative territories. In subcortical structures, the secondary gradient spanned from the thalamus and basal ganglia to the lateral geniculate nucleus and the cerebellum (Fig. 2c).

MP administration compressed the principal gradient, reducing the differentiation between unimodal and transmodal cortical systems while preserving the overall hierarchical organization observed during PL (Fig. 2a and b). This compression was asymmetric, with a more pronounced reduction in segregation in unimodal regions, reflected by elevated values in the somatomotor cortex and lower values in the DMN. In subcortical regions, MP shifted the principal gradient toward lower values characteristic of unimodal regions, further distinguishing the striatum, thalamus, and anterior cerebellum from the posterior cerebellar lobe and deep nuclei. In contrast, MP had no apparent effect on the secondary gradient (Fig. 2c and d).

**Low-dimensional connectivity gradients**
To quantitatively assess differences in macroscale cortical organization, we estimated the principal and secondary gradients for each individual MRI scan using diffusion embedding applied to parcellated

functional connectomes with 438 ROIs (Fig. 1d). Since the order and sign of diffusion embedding gradients vary across subjects, we computed 10 gradients, from which the two gradients that best aligned with the principal and secondary gradient templates (Fig. S1) were selected, separately for PL and MP. This approach allowed us to capture individual variations in hierarchical cortical organization while reducing computational complexity compared to dense connectomes. To statistically evaluate the effects of MP, we used linear mixed-effects (LME) models, including age, sex, race, body mass index (BMI), and intelligence quotient (IQ) as covariates.

There were systematic local shifts in the principal gradient across multiple cortical regions following MP administration (pFDR < 0.05; Fig. 3c and Table S1). Notably, sensory and motor areas, including the primary somatosensory (areas 1, 2, 3a, 3b) and motor cortices (area 4), exhibit a significant increase in gradient values (i.e., a shift towards less negative or more positive values; Cohen's d > 1.0, indicating a large effect size). This shift suggests a relative de-emphasis of unimodal processing in these regions. Conversely, higher-order cognitive areas, such as the prefrontal cortex (areas 46, a9-46d, p9-46v) and anterior insula (AVI, AAIC), show significant decreases in gradient values (−0.5 > Cohen's d > −1.0, indicating a medium-to-large effect size), indicating a relative reduction in their functional differentiation from

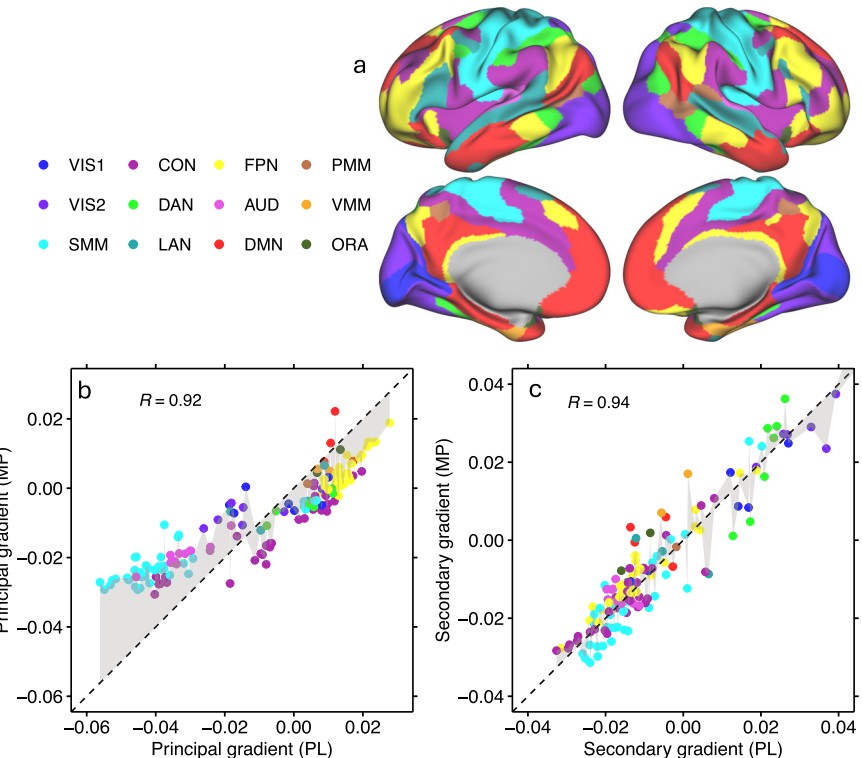

**Fig. 4 | Network-level organization and cross-condition consistency of cortical gradients. a** The 12 cortical network partitions of the iCSC-438 atlas overlaid on lateral and medial surface views of the left and right hemispheres. Scatter plots showing the strong linear association between principal (**a**) or secondary (**b**) gradient strengths in the placebo (PL) and methylphenidate (MP) conditions across 438 brain regions. In (**b**, **c**), data points are color-coded by network assignment[29]: Visual (VIS1, VIS2), Somatomotor (SMM), Cingulo-Opercular (CON), Dorsal Attention (DAN), Language (LAN), Frontoparietal (FPN), Auditory (AUD), Default-Mode (DMN), Posterior Multimodal (PMM), Ventral Multimodal (VMM), and Orbito-Affective (ORA). Diagonal lines indicate absence of MP-effects. Pearson correlations were tested using two-sided p-values. Source data are provided as a Source Data file.

primary sensory regions. This finding aligns with our hypothesis that MP enhances integration across cognitive domains by compressing the cortical gradient. Additionally, parietal and cingulate regions (e.g., area 24dd, IPS1, 7AL) also exhibit shifts consistent with increased integrative processing (−0.5 > Cohen's d > −1.0). The observed pattern suggests that MP administration compresses the hierarchical organization of cortical processing, reducing differentiation between unimodal and transmodal regions. This reconfiguration may facilitate greater cross-network integration and contribute to its cognitive-enhancing effects.

MP induced significant reductions in the principal gradient across several subcortical regions (−0.5 > Cohen's d > −1.0; pFDR < 0.05; Fig. 3c and Table S1). In the thalamus, both hemispheres showed reductions, particularly in regions involved in sensory and motor processing, such as the dorsal anterior medial and lateral nuclei[26]. MP also affected striatal regions, with reductions in the putamen and caudate (dorsal anterior and tail regions), which are linked to motor control and attention[27]. Changes in the globus pallidus and cerebellar regions (e.g., VI and VIIb) further suggest alterations in motor regulation and cognitive integration[28]. These findings reflect MP's impact on subcortical circuits involved in motor and cognitive functions. To a much lesser extent, MP also altered the secondary gradient in parietal, sensorimotor, primary visual cortex, prefrontal, temporal, and parietal regions (−0.49 > Cohen's d > −0.54, indicating a medium effect size; pFDR < 0.05; Fig. S2a–c; Table S1).

MP reduced excursions in the principal gradient, narrowing its range across 438 parcels (MP: [−0.03, 0.02]; PL: [−0.06, 0.02]) while maintaining a strong correlation with placebo values (R = 0.92; Fig. 4 and Fig. S2d). The statistical significance of MP-related changes (FDR-

corrected) was negatively correlated with the strength of the principal gradient during PL across regions, such that the reductions were stronger in areas with stronger principal gradient (R = −0.92; P < 1e-06, spin test). In contrast, the secondary gradient remained stable, exhibiting a similarly high correlation between MP and PL sessions (R = 0.94; Fig. 4 and Fig. S2d), with no appreciable reduction in its range or a clear association between regions of significant difference and gradient strength during PL. In the principal gradient, the MP-related attenuation of somatomotor (SMM) network segregation was more pronounced than in other functional networks[29], whereas no significant changes were observed in the secondary gradient in any network (Fig. 4b and c).

## Reproducibility

To evaluate the robustness of our findings, we conducted a reproducibility analysis using an independent dataset (Replication cohort; n = 20) acquired on a different group of healthy volunteers who were also tested twice; after 60 mg oral MP and after PL. All subjects were blinded to the drug conditions, and the sessions were fully counterbalanced to control for order effects[18,30]. There were no significant differences in FD between PL (0.240 ± 0.155 mm) and MP (0.261 ± 0.097 mm) conditions (P = 0.5, 2-sided paired t-test). In the Replication cohort, the effects of MP on low-dimensional connectivity gradients closely aligned with those observed in the Discovery cohort. Consistent with findings in the Discovery cohort, MP induced a compression of the principal gradient, reflecting a relative de-emphasis of unimodal processing and a reduced functional differentiation of these regions (0.6 < |Cohen's d | <1.2; Fig. 5). Notably, the secondary gradient remained largely unchanged, also consistent with findings in the

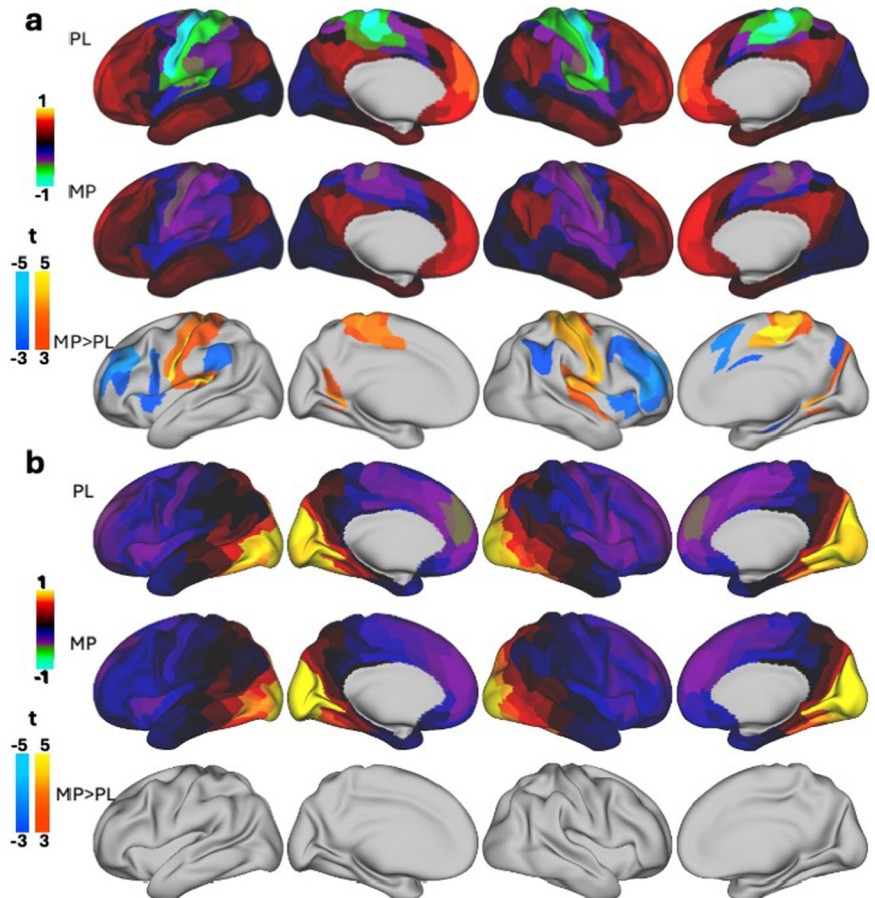

**Fig. 5 | Reproducibility of Methylphenidate's effects.** Surface renderings display the strength of the normalized principal (**a**) and secondary (**b**) gradients overlaid on lateral and medial views of the left and right hemispheres for the placebo (PL; top row) and 60 mg methylphenidate (MP; middle row) conditions in 20 healthy adults. The parcellation atlas included 438 cortical and subcortical regions. The bottom rows show the corresponding statistical difference maps (t-scores) estimated using a linear mixed-effects (LME) model with age, sex, and race as covariates, displayed using a false discovery rate (FDR) threshold of $p < 0.05$. All t-scores reflect two-sided tests.

Discovery cohort, indicating that MP's effects were more specific to the principal axis of cortical organization.

## MP and Visual attention

To explore a potential link between MP-induced changes in brain network organization and behavioral performance, we examined whether alterations in the strength of the principal gradient were associated with changes in visual attention (VA) accuracy, which, in addition to sustained attention, also demands working memory. Specifically, we tested whether individual differences in the MP-related compression of the principal gradient correlated with improvements in task accuracy across 32 participants who completed the ball tracking task inside the MRI scanner[31] (Fig. S3a). This exploratory analysis aimed to assess whether the reconfiguration of large-scale cortical hierarchy under MP might contribute to enhanced attentional performance.

VA accuracy (2- and 3-ball tracking runs averaged) was higher following MP administration compared to PL ($t = 2.6$, $P = 0.01$, paired t-test; Fig. S3b). Furthermore, individual increases in VA accuracy under MP were positively correlated with MP-induced changes in the strength of the principal gradient across multiple bilateral cortical regions ($P < 0.05$, uncorrected; Fig. S3c). This association was most pronounced in PF (supramarginal gyrus, inferior parietal cortex, Brodmann area 40; Fig. S3d), a region that plays a crucial role in allocating and maintaining focus on relevant visual stimuli during visual attention[32]. The association was also evident in other areas involved in spatial or visual attention (retrosplenial cortex, RSC[33], parieto-occipital sulcus area 2, POS2[34], peri-hippocampal area 3, PHA3[35], and temporal area 1 posterior, TE1p[36]), and in the right posterior angular gyrus (PGp), a core DMN node in the inferior parietal cortex.

## Dopamine D1 and D2/3 receptor availability

To quantify D1R and D2R availability and assess the effects of MP on dopamine in the striatum, we computed non-displaceable binding potential (BPnd) from scans with [11C]NNC-112 (measured at baseline) and with [11C]raclopride (measured after placebo and after MP) using the cerebellum as a reference region[37–39].

As expected, raclopride PET data revealed regionally specific patterns of D2 receptor binding (Fig. S4), with MP administration producing significant reductions in BPnd across the putamen, caudate, and nucleus accumbens ($P < 0.0001$, two-sided; Fig. 6a and Supplementary Fig. S5) reflecting the competition for receptor binding from enhanced levels of dopamine. Across participants, D1R and D2R availability, as well as MP-induced DA increases in the striatum, exhibited substantial interindividual variability and approximated normal distributions ($W > 0.94$, $P = 0.05$, Shapiro-Wilk test). No significant association was observed between D1R availability and DA increases. However, D2R availability in the striatum positively correlated with DA levels under placebo and negatively under MP ($|R| > 0.44$, $P < 0.008$; Fig. S6).

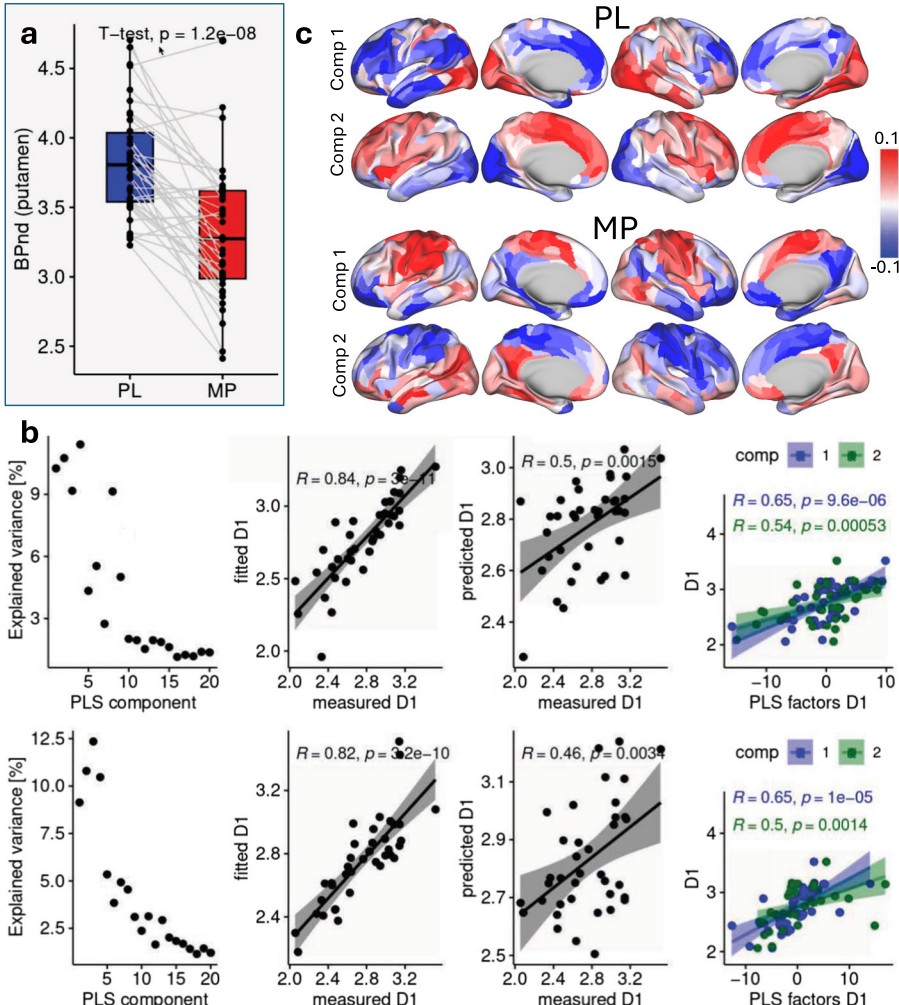

**Fig. 6 | Resting-state functional gradients predict striatal dopamine D1 receptor availability. a** Paired plot showing the non-displaceable binding potential (BPnd) of [$^{11}$C]raclopride in putamen for methylphenidate (MP) and placebo (PL) sessions across 38 healthy adults. Each box shows the median (center line), the interquartile range (IQR; bounds of the box representing the 25th and 75th percentiles), and the minimum and maximum values within 1.5 times the IQR (whiskers); individual lines connecting the boxes represent paired measurements from the same subject. **b** Partial least squares (PLS) regression predicting D1 receptor availability in the putamen from the secondary gradient across 438 brain parcels with 2 components. The top row includes: (i) explained variance by PLS components, (ii) fitted values from leave-one-out cross-validation (LOO-CV) versus measured values for the PL session, (iii) predicted D1 receptor availability in MP using the PLS model trained on PL, and (iv) the relationship between the measured metric and PLS factors for components 1 and 2. The shaded area around the regression line represents the 95% confidence interval. The bottom row shows the same metrics but with the model trained on MP and tested on PL. **c** Loading patterns for components 1 and 2 of the PLS model predicting D1 receptor availability from the secondary gradient corresponding to PL and MP sessions. Surface renderings depict lateral and medial views. All Pearson correlation tests are two-sided. Source data are provided as a Source Data file.

## FC gradients predict dopaminergic function

To examine the relationship between macroscale cortical organization and dopaminergic function, we applied PLS regression with leave-one-out cross-validation (LOO-CV) to predict D1R and D2R availability as well as MP-induced DA increases in striatum, independently from the principal and secondary gradients (Fig. 1f). First, we trained a PLS model with 2 components using the gradient maps of the PL sessions and tested its ability to predict dopaminergic measures using the gradient maps of the MP sessions. To ensure the robustness of our findings, we then reversed the order of training and testing, using the MP gradients to train the model and testing it using the PL gradients. This approach allowed us to assess whether the relationship between cortical hierarchy and striatal DA measures remained stable across pharmacological conditions.

While the principal gradient predicted striatal D1R, D2R, and DA increases when the model was trained and tested within the same condition (PL or MP), it failed to generalize across conditions (i.e., PL to MP or MP to PL; Figs. S7–S11). This likely reflects the substantial differences in the principal gradient between the PL and MP states. In contrast, the secondary gradient remained relatively stable across conditions. This gradient consistently predicted D1R availability in the putamen and nucleus accumbens, explaining up to 37% of the variance in D1R in the nucleus accumbens (Fig. 6b and Fig. S12, third column). In within-condition training (PL or MP), the two PLS components explained over 20% of the variance, and models based on the secondary gradient accurately fit dopaminergic metrics ($R > 0.8$; $P < 1$e-10). Moreover, individual PLS components derived from the secondary gradient were significantly associated with D1R availability in bilateral putamen and nucleus accumbens ($R > 0.5$; $P < 0.001$; Fig. 6b and Fig. S12), suggesting that interindividual differences in the secondary gradient reflect underlying variability in D1R availability.

The secondary gradient of the Test sessions (MP/PL), which was not seen by the model during training, predicted D1R availability in bilateral putamen and nucleus accumbens ($R > 0.46$; $P = 0.003$), and

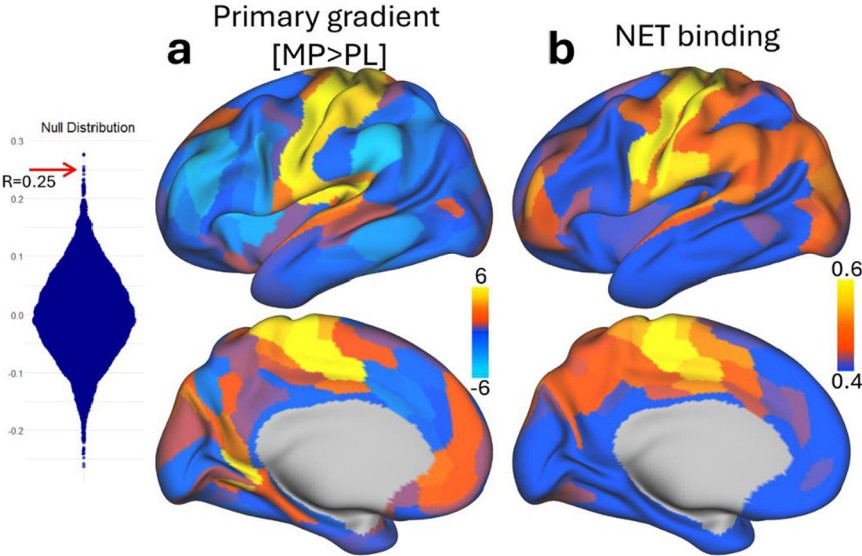

**Fig. 7 | Association between principal gradient shifts and NET density. a** Cortical surface maps showing the statistical differences (t-score) in the principal functional connectivity gradient between methylphenidate (MP) and placebo (PL), displayed on lateral (top) and medial views of the left hemisphere. **b** Regional norepinephrine transporter (NET) density map derived from PET data, displayed on corresponding cortical views. The null distribution of spatial correlation coefficients was generated via spin test (10,000 permutations). The observed empirical correlation between principal gradient shifts and NET density across 438 parcels is indicated by a red horizontal arrow. A significant correlation was found (spin-test $p = 0.0008$), indicating that regions with higher NET density had stronger MP-induced shifts in the principal gradient. All statistical tests are two-sided.

the loadings of the PLS components (Fig. 6c and S12) were highly correlated with the principal gradient ($R > 0.3$; $P < 1e-06$, spin test) and to a lesser degree with the secondary gradient. The secondary gradient also predicted D2R availability in caudate, explaining more than 15% of D2R variance in the bilateral caudate during the MP session but not during the PL session ($R > 0.38$; $P < 0.02$; Fig. S13). However, the predictions of D1R in caudate, and D2R and DA increases in striatum from the secondary gradient were inconsistent across Test sessions (MP/PL; Fig. S14–S17).

### Gradient shifts and regional DAT and norepinephrine transporter (NET) density

Since MP block both DAT and NET we tested whether principal gradient shifts were most prominent in brain regions enriched for MP's primary pharmacological targets: DAT and NET. Specifically, we tested the spatial correspondence between statistical MP–related differences in the principal gradient and publicly available PET-derived maps of DAT and NET density. Spatial correlations were assessed using a spin test to account for spatial autocorrelation. We found a significant correlation between NET density and MP-induced shifts in the principal gradient ($r = 0.25$; spin-test $p = 0.0008$; Fig. 7); such that regions with higher NET density exhibited larger shifts in the principal gradient with MP. No significant correlation was observed between DAT density and the principal gradient ($r = 0.13$; spin-test $p = 0.09$). These results suggest that the regional expression of NET but not DAT mediates MP-related modulation of principal gradient organization.

### Validation of gradient compression in medicated children from the ABCD study

To test whether our findings generalize, we analyzed rsfMRI data from the Adolescent Brain Cognitive Development (ABCD) study using the same low-dimensional diffusion embedding approach used for the Discovery and Replication samples. We compared the principal gradient of cortical organization in children with ADHD who were taking stimulant medications (amphetamine or methylphenidate) ($n = 379$) to unmedicated children ($n = 4579$). FD was slightly higher for medicated ($0.120 \pm 0.042$ mm) than for unmedicated ($0.115 \pm 0.042$ mm) children ($P = 0.012$, 2-sided t-test). We hypothesized that medicated children would show reduced differentiation along the principal gradient, like the gradient compression seen in adults after MP. A factorial ANCOVA controlling for inattention, sex, age, head motion, scanner type, and research site revealed that medicated children had significantly higher gradient values in primary somatomotor areas (1, 3a, 4) and posterior insula, and lower values in the inferior frontal cortex than unmedicated peers ($0.16 < |\text{Cohen's d}| < 0.22$, indicating a small effect size; $p < 0.001$, uncorrected; Fig. 8), which were more pronounced in girls than boys (Fig. S18). These patterns mirror the MP-related gradient compression seen in our adult cohorts, suggesting that stimulant treatment may also promote a more integrated cortical organization in children.

### Discussion

Here, we show that MP administration altered macroscale cortical organization by compressing the principal gradient while preserving its overall hierarchical structure. This aligns with the hypothesized role of MP in enhancing cognitive function by integrating functional connectivity across cortical and subcortical networks. Additionally, while the principal gradient was significantly altered by MP, the secondary gradient remained largely stable, suggesting that MP preferentially affected the hierarchical axis spanning from unimodal sensory to transmodal association areas, rather than the gradient spanning anterior to posterior neocortex. These findings were reproduced in independent Discovery and Replication cohorts.

Under PL, the principal gradient showed the expected hierarchical organization, with sensory and motor regions exhibiting the lowest values and higher-order association areas (DMN and FPN) showing the highest values, consistent with prior work[15]. MP administration compressed this gradient in both unimodal and transmodal areas, suggesting decreased segregation and enhanced integration across cortical domains. The MP-induced gradient shifts involving the interior

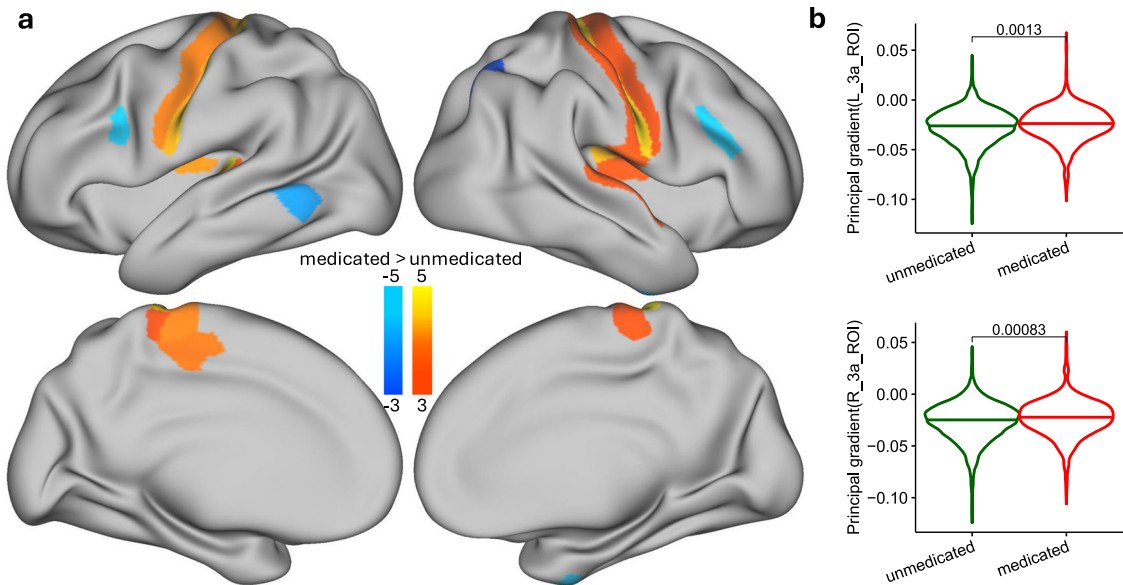

**Fig. 8 | External validation of stimulant-related gradient compression in children.** Children from the ABCD study diagnosed with ADHD and treated with stimulant medications (methylphenidate or amphetamines; 105 girls, 274 boys) were compared to unmedicated peers (2260 girls, 2319 boys). **a** Surface maps show uncorrected group differences (ANCOVA, $p < 0.001$) in the principal cortical gradient between medicated and unmedicated children, adjusting for inattention, sex, age, head motion, scanner type, and study site. The pattern of differences, displayed on ventral and lateral views of both hemispheres, mirrors the stimulant-related gradient compression seen in adults. All t-tests are two-sided. **b** Violin plots depict significantly higher gradient values in medicated children within bilateral anterior primary somatomotor cortex (area 3a), after correcting for inattention, sex, age, head motion, scanner type, and research site; P-values are from two-sided two-sample t-test. Source data are provided as a Source Data file.

frontal cortex and basal ganglia align with task-based fMRI evidence that MP increases activation in these regions to enhance cognitive control in ADHD[40]. Although MP compressed the principal gradient, indicating reduced cortical segregation across the unimodal-transmodal distinction, this does not preclude strengthening of the negative correlation between default-mode and task-positive networks, both of which are networks comprised mostly of transmodal cortices. Although neither the primary nor the secondary gradients captured this negative connectivity, it is possible that one of the additional gradients might reflected it but further work is needed to clarify this. Thus, while we interpret our findings to reflect an enhanced integration across unimodal and transmodal cortices it could concurrently occur with more coordinated task-specific network interactions as reported by prior task-based fMRI studies in ADHD[41].

Pedersen et al.[13], linked functional gradients to D1R distribution and our findings are consistent with their results while further expanding the understanding of dopamine's modulatory influence on these gradients. While their work provided an important anatomical link between functional hierarchy and dopaminergic systems, their analysis was cross-sectional and did not examine how pharmacological manipulations could dynamically reshape gradient architecture. Our results build on this by showing that MP-induced modulation of the principal gradient is functionally meaningful. It predicts PET-measured DA responses to MP in both cortical and subcortical regions, as well as baseline D1R and D2R availability. This supports the idea that functional gradients are not static but can be compressed by neuromodulatory input, particularly dopaminergic stimulation.

Subcortical gradient organization was also affected by MP, particularly in the principal gradient, which further differentiated medial and lateral subdivisions of the thalamus, dorsal and ventral striatum, and anterior/posterior cerebellum compared to PL. MP compressed this gradient, reducing the functional differentiation across subcortical subregions; however, in contrast to the cortical gradient, it shifted them preferentially toward unimodal partitions. MP effects in

subcortical regions is consistent with prior work demonstrating dopaminergic influences on thalamic, cerebellar, and striatal integration[42–44].

The exploratory analysis revealed that MP-induced changes in cortical gradients were associated with individual differences in visual attention performance. Specifically, participants showing greater compression of the principal gradient under MP (i.e., reduced segregation between sensory and associative areas, which we interpret as a mechanism supporting MP's effects on attention and cognition) also exhibited stronger task-related improvements in visual attention. These findings align with previous work suggesting that MP modulates large-scale brain network organization to facilitate cognitive processes such as attention[45–47]. MP has been shown to enhance attention[48], a finding that is consistent with the broader literature on its effects on cognitive function[49], particularly in attention-related tasks[50]. The association between MP-induced changes in visual attention and shifts in the principal gradient was most pronounced in areas critically involved in sustained or visual attention[32–36,51]. This finding is consistent with the hypothesis that a pharmacological manipulation, like MP, can enhance attentional performance by increasing the integration of brain regions involved in higher-order cognitive functions[52].

In contrast to the principal gradient, the secondary gradient remained largely stable across PL and MP conditions. Yet its individual variation carried important neurochemical information: higher D1R availability in the putamen and nucleus accumbens and D2R availability in the caudate were significantly associated with the secondary gradient. These findings align with the known distribution and functions of these receptors; D1R facilitates goal-directed behavior and reward learning[53–55], while D2R modulates habit formation and inhibitory control[56–58]. Interestingly, while the principal gradient was more sensitive to MP-induced reorganization, the secondary gradient appears to reflect stable, trait-like patterns of functional connectivity that are shaped by underlying dopamine receptor distributions. These patterns are likely shaped by intrinsic factors, such as the spatial

distribution of dopamine receptor subtypes (e.g., D1R, D2R) across brain regions, which vary across individuals and may influence baseline connectivity architecture. Thus, whereas the principal gradient indexes state-dependent reconfigurations linked to transient neuromodulatory effects, the secondary gradient may encode more enduring, receptor-driven properties of brain organization that explain interindividual differences in cognitive function and behavior, even in absence of a pharmacological challenge. Although MP-induced changes in the principal gradient reveal state-dependent reorganization, correlations between baseline D1R/D2R availability and the secondary gradient suggest that this gradient reflects more enduring, receptor-driven aspects of functional brain organization. Indeed, we recently demonstrated that the ratio of striatal D1R to D2R availability predicts individual variability in attention-related neural responses to methylphenidate[31], underscoring the role of dopamine receptor distribution in shaping stable brain-behavior relationships.

Additionally, we found that MP-induced regional changes in the principal gradient were predicted by regional NET density, consistent with MP's pharmacological profile as a dual DAT and NET blocker. This is further supported by the absence of an association with cortical DAT density, likely reflecting the low expression of DAT in the cortex[59]. In cortical regions, NETs mediate dopamine reuptake[60], and their blockade increases extracellular dopamine levels[61]. These findings suggest that cortical gradient modulation under MP is influenced more by NET than DAT, aligning with prior evidence that NET-rich areas contribute to MP-induced alterations in brain network topology[62]. Future studies are warranted to further explore the role of norepinephrine in shaping cortical gradient dynamics.

Our findings align with the framework proposed by Shine and colleagues[63,64], which emphasizes the role of neuromodulatory systems in shaping large-scale neural dynamics through alterations in cortical circuit properties. By enhancing neuromodulatory tone, MP may recalibrate the balance between integration and segregation across cortical hierarchies, effectively reconfiguring large-scale network topology in a manner that facilitates adaptive cognitive control. These findings support the idea that DA and other neurochemical systems shape hierarchical brain organization and macroscale functional gradients[65]. The shift in the principal gradient under MP likely reflects dopaminergic modulation along this chemoarchitectural axis, aligning with the flexible reconfiguration reported under enhanced NE transmission[62], and the known effects of psychostimulants on attention-related systems[66]. Recent evidence from ADHD further demonstrates that cortical gradient perturbations co-localize with multiple neurotransmitter systems and gene expression patterns related to specific cell types[67], reinforcing the link between neuromodulatory influences, cortical gradient architecture, and individual differences in brain function.

Our analysis of the ABCD dataset provides compelling external validation of our primary findings by demonstrating that children with ADHD who were receiving stimulant medications exhibit cortical gradient alterations consistent with those observed in adults following MP administration. Medicated children showed higher gradient values in somatomotor regions, and lower values in the inferior frontal cortex, mirroring the spatial pattern of gradient compression identified in our adult cohorts. These results suggest that stimulant treatment may shift cortical organization toward a more integrated transmodal configuration even during neurodevelopment. Importantly, these differences remained significant after accounting for inattention severity and sex, indicating that the observed effects are likely attributable to pharmacological intervention rather than baseline behavioral differences. However, effect sizes were attenuated compared with adults, likely reflecting developmental differences, heterogeneity in doses and medication type (MP and other stimulants), and the cross-sectional design of the ABCD dataset. Despite these limitations, the convergence of findings across age groups supports the hypothesis that stimulant medications exert a consistent, system-level influence on the macroscale organization of the cortex, potentially enhancing communication between sensorimotor and transmodal regions critical for attention and executive function.

Limitations of our study include the focus on striatal measures of D1R and D2R availability; future research should investigate the role of cortical D1R and D2R in shaping the principal gradient. Additionally, a limitation of the ABCD dataset is the inability to confirm whether children had taken their prescribed stimulant medication on the day of the scan.

In conclusion, our study demonstrates that MP selectively reshapes macroscale functional gradients, particularly by enhancing cortical integration and refining subcortical distinctions. These changes are closely linked to dopaminergic function, with the secondary gradient emerging as a key predictor of individual variations in DA receptor availability. MP-induced shifts in the principal gradient were associated with improvements in VA accuracy and the regional distribution of NET in the cortex. These findings provide a mechanistic framework for understanding MP's effects on brain functional organization and highlight the utility of functional gradients in investigating neuropharmacological interventions.

## Methods
### Participants
The *Discovery* cohort consisted of 38 healthy adults (ages 22–64 years; 14 females), while the *Replication* cohort included 20 healthy adults (mean age: 22–56 years; 9 females)[18,30]. All participants provided written informed consent for their participation in the study, which was approved by the Institutional Review Board (IRB) at the National Institutes of Health (NIH). Exclusion criteria encompassed a history of substance abuse or dependence (excluding nicotine), psychiatric disorders, neurological diseases, medical conditions that could impact cerebral function (including cardiovascular, endocrine, oncological, or autoimmune disorders), current use of prescribed or over-the-counter medications, and any history of head trauma involving loss of consciousness exceeding 30 min. Some PET and task-fMRI data from these samples have been analyzed in previous publications using different methods[18,30,31,68–73]. However, the analyses and findings presented in this paper are entirely original.

Participants in the Discovery cohort completed two imaging sessions on separate days, one week apart, each conducted under either 60 mg oral dose of MP or PL in a randomized order and counterbalanced (Fig. 1a and b). PET scans were first performed to assess D1R and D2R availability, followed by MRI sessions in which rsfMRI signals (21 min total across three consecutive scans) and task-based fMRI during a VA task (12 min across two runs) were collected. Findings from brain activation during the VA task have been previously reported[31]. In the present work, we focus specifically on how MP modulates the relationship between changes in VA accuracy and changes in the strength of the principal gradient.

Participants in the Replication cohort completed three imaging sessions spaced an average of $40 \pm 35$ days apart[18,30]. During each session, participants received one of three pharmacological conditions: (1) oral MP (60 mg) with intravenous (IV) placebo (3 cc saline), (2) oral placebo with IV-MP (0.25 mg/kg in 3 cc sterile water), and (3) oral and IV placebo. Session order was randomized and counterbalanced across participants to ensure a double-blind, placebo-controlled design. The IV-MP data were not included in the present analyses, which focus exclusively on oral MP.

For external validation we analyzed rsfMRI data from the Adolescent Brain Cognitive Development (ABCD) study. Our sample included 4579 unmedicated children (2260 girls and 2319 boys) and 379 children (105 girls and 274 boys) receiving stimulant medications for ADHD (methylphenidate or amphetamines) with available inattention scores (see below). Only participants without medical or

cognitive impairments, limited English proficiency, or MRI contra-indications were included[74]. The ABCD study was approved by the institutional review board at the University of California, San Diego, and by 21 data collection sites across the United States[75,76].

## Inattention scores

For statistical analysis of the ABCD data we obtained the Attention Problems subscale of the Child Behavior Checklist (CBCL) from the NIMH Data Archive (https://nda.nih.gov). This parent-reported scale includes 10 items rated on a 3-point scale (0 = not true, 1 = somewhat true, 2 = very true) that reflect common attention-related behaviors, such as difficulty concentrating, restlessness, impulsivity, and poor school performance. The total inattention score was calculated by summing the ratings across all 10 items, yielding a continuous score ranging from 0 to 20[77].

## Visual attention

Thirty-two participants from the Discovery sample completed a visually based, non-verbal, VA task inside the scanner[48] (Fig. 5a). Participants were instructed to covertly follow two or three designated target balls out of a set of ten. In each 12-second trial, the target balls were briefly highlighted for 0.5 s, followed by Brownian movement of all balls across the screen for one second. Participants tracked the targets while fixating on a central cross. After Brownian movement ceased, a new set of balls was briefly highlighted, prompting participants to respond via button press with the right index finger if the highlighted balls matched the original targets. A 1-second response window was provided, after which the correct target set was re-highlighted for 0.5 s to reorient attention before the next trial began. Each participant completed two runs, one involving tracking two balls and the other three balls, with each run including 15 ball-tracking trials (3 min). Task stimuli were presented on a BOLDscreen 32 liquid-crystal display (Cambridge Research Systems, UK).

## PET image acquisition and processing

For participants in the Discovery cohort, PET imaging was conducted to assess dopamine receptor availability with two radiotracers: [11C]NNC-112 for D1R and [11C]raclopride for D2R. Scans were performed using one of two systems: a high-resolution research tomography (HRRT) scanner (n = 17; Siemens AG, Germany) or a Biograph PET/CT scanner (n = 21; Siemens AG, Germany). The use of multiple scanners was necessitated by scheduling constraints; methods for harmonizing data across systems are detailed in the PET analysis section. [11C]NNC-112 scans were conducted at 10 AM under baseline conditions, without pharmacological intervention. [11C]raclopride scans were conducted on two separate days: once following the administration of an oral PL to establish baseline D2R availability and once 60 mg MP (single-blind; counterbalanced session order). To maintain consistency, [11C]raclopride scans were always performed at 1 PM on the same scanner for each participant.

For [11C]NNC-112, emission imaging began immediately after an injection of up to 15 mCi, with 21 dynamic frames collected over 90 min post-injection. For [11C]raclopride, emission imaging commenced immediately following an injection of up to 10 mCi, capturing 22 dynamic frames over 60 min. All dynamic images were reviewed before analysis to exclude data compromised by motion artifacts or misalignment.

FreeSurfer v5.3.0 (http://surfer.nmr.mgh.harvard.edu) was used for automatic segmentation of anatomical MRI scans based on the automated segmentation (ASEG) atlas. This process identified sub-cortical regions of interest, including the bilateral nucleus accumbens, caudate, and putamen as well as the cerebellum. Time–activity curves from striatal voxels were used to calculate BPnd, the ratio of specifically bound ligand to the non-displaceable (free + nonspecifically bound) ligand in the brain, using the simplified reference tissue

model[39], and distribution volume ratios (DVR) based on the Logan reference tissue model[37,38], using the cerebellum as a reference region. The resulting striatal voxel-to-cerebellum DVR values correspond to BPnd+1, which was used to confirm D1R and D2R receptor availability. BPnd images were co-registered with high-resolution T1- and T2-weighted structural MRI scans. MAGIA (Metabolic and Anatomical imaging Analysis), an automated Matlab pipeline for PET image processing that integrates preprocessing, kinetic modeling, and region-based quantification, was used for this purpose[78]. We used an updated version of the ComBat Harmonization technique[79], which operates within an empirical Bayes framework, due to its superiority over other harmonization methods[80,81]. Harmonization across PET cameras was performed separately for each tracer and PET-derived receptor availability measure.

## MRI acquisition

Participants in the Discovery cohort underwent structural and rsfMRI in a 3.0 T Magnetom Prisma scanner (Siemens Medical Solutions USA, Inc., Malvern, PA) equipped with a 32-channel head coil twice. rsfMRI data were acquired using a multi-echo, multiband echo-planar imaging (EPI) sequence[82] with a multiband factor of 3 and anterior-to-posterior phase encoding. The imaging parameters included a repetition time (TR) of 891 ms, echo times of 16, 33, and 48 ms, a flip angle of 57 degrees, and 45 slices with a voxel size of 2.9 × 2.9 × 3.0 mm. To ensure the stability of the functional connectivity (see below), a total of 471 volumes were collected over a 7-min scan[83] while participants remained awake with their eyes open. All participants were provided with earplugs to reduce the influence of the scanner's acoustic noise[84] on brain activity[85]. A fixation cross was displayed on a black background via a liquid-crystal display screen (BOLDscreen 32, Cambridge Research Systems, UK) in a dimly lit room. High-resolution anatomical images were obtained using a 3D magnetization-prepared rapid gradient echo (MP-RAGE) sequence[86] (TR = 2400 ms, TE = 2.24 ms, flip angle = 8 degrees) and a variable flip angle turbo spin-echo sequence[87] (Siemens SPACE; TR = 3200 ms, TE = 564 ms). The anatomical scans had an isotropic voxel size of 0.8 mm, with a field of view (FOV) of 240 × 256 mm, a matrix size of 300 × 320, and 208 sagittal slices.

## Simultaneous PET/fMRI acquisition

Participants in the Replication cohort underwent simultaneous PET/MRI data in a 3 T Siemens Biograph mMR scanner (Medical Solutions, Erlangen, Germany) equipped with a 12-channel head coil. The simultaneous PET/fMRI protocol is detailed elsewhere[18]. Briefly, imaging lasted 90 min, with synchronized PET and fMRI acquisition enabled by timestamped list-mode PET data. PET acquisition began 30 min after oral pill administration, following a manual bolus injection of [11C]raclopride (15.7 ± 1.9 mCi; injection duration: 5–10 s). fMRI data were continuously collected using a whole-brain single-shot gradient-echo EPI sequence (TR/TE = 3000/30 ms; flip angle = 70°; matrix = 64; 36 axial slices; 4 mm slice thickness; 3 mm in-plane resolution; 1800 volumes). At 30 min post-radiotracer injection, participants received an intravenous injection of either PL (saline) or MP (0.25 mg/kg) over ~30 s. Throughout the 90-min scan, participants were instructed to relax with their eyes open and remain as still as possible. In this study, we used rsfMRI data collected after intravenous injection (60-min; 1200 volumes per scan) for both the PL and oral MP conditions. We excluded the initial 30 min of rsfMRI data (prior to the IV injection) to focus on the period during which the pharmacological effects of intravenous MP or PL were expected to manifest most strongly (1–2 h).

## ABCD imaging procedures

Imaging in the ABCD study was conducted on 3 T MRI scanners (Siemens Prisma, Philips, and GE 750) using standardized protocols across 21 sites. All scanners were equipped with adult-sized multi-channel coils and supported multiband echo planar imaging (EPI). Structural

MRI included high-resolution 1 mm isotropic T1-weighted and T2-weighted scans. Functional MRI data were collected using T2-weighted multiband EPI with a TR of 800 ms, TE of 30 ms, 2.4 mm isotropic resolution, 60 whole-brain slices, a flip angle of 52°, and a multiband acceleration factor of 6. Additional technical details are available elsewhere[88,89].

## MRI processing

Resting-state functional MRI data were preprocessed using fMRIPrep[90] (version 24.0.1). Anatomical preprocessing included skull stripping, bias field correction, segmentation, and nonlinear spatial normalization to the MNI152NLin2009cAsym template. Functional images underwent motion correction, slice timing correction, and realignment to the mean functional image. Spatial normalization was performed using nonlinear warping. The TE-dependent analysis (tedana), an open-source Python tool designed for processing multi-echo fMRI data[91], was utilized to produce an adaptive T2* map and compute an optimally weighted combination of single-echo time series. Both volume-based and surface-based preprocessing were performed. Functional time series were resampled to the subject's cortical surface using the FreeSurfer-derived white and pial surfaces and registered to the fsLR and fsaverage space for group-level analysis. For subsequent analysis, data were processed in 32k CIFTI format with 32,492 cortical vertices per hemisphere for a total of 91,282 grayordinates[92], which allows for high-resolution representation of both cortical and subcortical structures. Physiological and motion-related noise sources were identified and removed with nuisance regressors including global signals, white matter and cerebrospinal fluid signals and their temporal derivatives, temporal component-based physiological noise correction (tCompCor), six motion parameters (3 translations and 3 rotations), their temporal derivatives, and high-order polynomial trends. Framewise displacement (FD) and the root mean square (RMS) of voxelwise intensity differences between successive time points (e.g., DVAR) were computed, and treated as nuisance regressors, and volumes exceeding 0.5 mm FD and 0.5% DVARS were flagged to facilitate posterior denoising based on scrubbing[93]. The final preprocessed images were resampled to 2 mm isotropic resolution in MNI space. Quality control metrics, including head motion, signal-to-noise ratio, and registration accuracy, were visually inspected to ensure preprocessing integrity.

Our external validation analyses used data from the ABCD Brain Imaging Data Structure Community Collection (ABCC; https://collection3165.readthedocs.io/en/stable/), which provides high-quality resting-state fMRI acquired over 20 min. Preprocessing followed the ABCD-BIDS pipeline, which performs brain extraction, denoising, and normalization of structural images, cortical and subcortical segmentation and surface reconstruction with FreeSurfer[89] for pediatric use[94], conversion to HCP-compatible CIFTI format, alignment of fMRI time series to standard space, projection of fMRI data to cortical surfaces, and a motion correction method that accounts for respiratory artifacts[95] and standard denoising, regressing out head motion, white matter, CSF, and global signals that could confound group comparisons[96,97].

## Diffusion embedding

We used the connectivity gradient approach based on diffusion embedding[15] to map the principal functional connectivity gradients with the python package mapalign (https://github.com/sensein/mapalign). Specifically, to extract high-dimensional gradients we computed "dense connectomes" (e.g., Pearson correlation matrices with 91,282 × 9,1282 edges) in Matlab 2022a, and corresponding adjacency matrices retaining the top 10% of positive edges per row in Python 3.9.15. Then we constructed similarity matrices based on cosine distance, which were used as inputs for data driven diffusion embedding to identify 2 principal diffusion components based on Markov transition probabilities with $\alpha = 0.5$ and eigenvector decomposition[15] in Python.

Since extracting network organization from dense connectomes using the diffusion embedding technique[15] demands significant computational power, often exceeding the capacity of standard computing systems, we optimized the process by representing connectivity profiles as "parcellated connectomes" (e.g., Pearson correlation matrices between a set of ROIs) rather than computing pairwise connectivity for every grayordinate. This low-dimensional approach substantially reduces computational complexity while preserving essential connectivity information[14], in line with prior research[98]. The 360 partitions of the multi-modal parcellation of the human cerebral cortex[99], the 50 thalamic and basal ganglia partitions of the Melbourne Subcortex Atlas[100], and the 28 cerebellar partitions of a probabilistic atlas of the human cerebellum[101] were combined in the Integrated Cortico-Subcortical-Cerebellar Atlas (iCSC-438), a dense parcellation with 438 partitions organized into 12 canonical network partitions[29] in CIFTI space, and used to compute individual functional connectomes for each scan. Specifically, a 438 × 438 correlation matrix was computed for each individual scan using the Connectome Workbench (https://www.humanconnectome.org/software/get-connectome-workbench). Then, an adjacency matrix with only the top 10% strongest positive connections per row was created and used to compute a similarity matrix based on cosine distance, and diffusion embedding was used to extract 10 low-dimensional connectivity gradients per scan using the mapalign Python package.

Gradient mapping from dimensionality reduction algorithms can vary across subjects due to differences in component order and sign. To ensure comparability, we aligned individual gradients to a standardized group-level template. For this purpose, we created group-level templates for the principal and secondary gradients from the average parcellated connectome, applying the same procedure used for individual scans. Then, for each individual scan, the components with the highest spatial correlation to these templates were selected as the principal and secondary gradients and used in group-level statistical analyses.

## Correlations with neurotransmitter transporter expression

We tested whether FC gradient shifts were most prominent in brain regions enriched for MP's primary pharmacological targets: DAT and NET. Specifically, we assessed spatial correlations between the MP-induced shift in FC gradients and publicly available maps of DAT and NET density derived from prior PET studies, as implemented in Python's neuromaps package[102]. Simple spatial correlations between brain maps are inadequate due to spatial autocorrelation (nearby cortical vertices exhibit similar signal patterns, violating the assumption of independence and inflating correlation coefficients). To address this, we used a spin test that generates null distributions by applying random rotations to spherical projections of cortical data, preserving the spatial autocorrelation structure while randomizing regional assignments[103]. We computed the spatial correlations between the maps of statistical MP–PL differences in the principal gradient and the PET-derived DAT and NET maps. The null distribution consisted of 10,000 permutations of the gradient difference maps. Correlations were considered significant if they exceeded the 95th percentile of the null distribution.

## Statistical analysis

To assess the effect of MP on the strength of the connectivity gradients compared to PL, we implemented an LME modeling approach in Matlab. Separate LME models were fit for each gradient, with fixed effects for drug condition. Age, sex, race, BMI, and IQ were included as covariates to control for potential confounding effects. A random intercept for each participant was included to account for within-subject dependencies across the three resting-state scans collected in

each drug condition, which were acquired in a randomized order. Model selection and inference were based on likelihood ratio tests and restricted maximum likelihood estimation. False discovery rate (FDR) correction was applied to account for multiple comparisons. To assess the main effect of ADHD stimulant medications (MP and amphetamines) on the principal gradient in the ABCD sample we conducted a factorial analysis of covariance (ANCOVA) in MATLAB using inattention, age, and head motion, as a continuous covariates and sex, research site, and scanner type as categorical covariates.

### PLS analysis

Partial Least Squares (PLS) regression was performed in R using the pls package to predict DA metrics from low-dimensional gradients. The predictor (gradient components) and outcome (D1R, D2R, DA increases) variables were z-scored to ensure comparability. The optimal number of PLS components was determined using cross-validation, selecting the model that minimized prediction error. Model performance was evaluated using $R^2$ and root mean squared error (RMSE).

### Reporting summary

Further information on research design is available in the Nature Portfolio Reporting Summary linked to this article.

## Data availability

The deidentified ROI summary data generated in this study is available in Figshare (https://doi.org/10.6084/m9.figshare.30433099). Deidentified individual-level images can be obtained from the corresponding author upon reasonable request. Per NIH policy, sharing requires Data Transfer Agreements between institutions to ensure compliance with privacy and institutional requirements. ABCD data are publicly available through the NIMH Data Archive (https://data-archive.nimh.nih.gov/abcd). Both Individual ROI and group-averaged imaging data generated in this study are provided in the Source Data file. Source data are provided with this paper.

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

## Acknowledgements

We are thankful to Adam Thomas, PhD, Dustin Moraczewski, PhD, and Eric Earl, BS (National Institute of Mental Health Data Science and Sharing Team) for providing access to the ABCD Community MRI Collection (NDA collection 3165) data on our servers. This study utilized the computational resources of the NIH HPC Biowulf cluster (http://hpc.nih.gov). This work was done with support from the National Institute on Alcohol Abuse and Alcoholism (Y1AA-3009; ZIAAA000550). Part of the data used in this study were obtained from the Adolescent Brain Cognitive Development (ABCD) Study (https://abcdstudy.org/) and are stored in the NIMH Data Archive. The ABCD Study is supported by the National Institutes of Health (NIH) and its affiliated federal partners. ABCD consortium investigators provided data but did not participate in the analysis or writing of this report. The content of this manuscript is solely the responsibility of the authors and does not necessarily represent the official views of the NIH or the ABCD consortium.

## Author contributions

D.T. and N.D.V. conceived and designed the study. D.T. and P.M. developed the neuroimaging protocol and supervised data acquisition. D.T., P.M., S.B.D., W.Y., K.B.M., F.V., J.Z., C.L., M.V.Y., S.A., M.V., and J.G.W. collected or preprocessed the PET and MRI data. D.T. performed the data analysis. D.T. and N.D.V. interpreted the results and drafted the manuscript. All authors contributed to the interpretation of findings, revised the manuscript critically for important intellectual content, and approved the final version of the paper.

## Funding

## Competing interests

The authors declare no competing interests.
