## [Transparent Peer Review file · Nature Communications]

Methylphenidate reorganizes cortical hierarchy through dopaminergic modulation

Corresponding Author: Dr Dardo Tomasi

Version 0:

Reviewer comments:

Reviewer #1

(Remarks to the Author)

This is a well conducted combined fMRI/PET study that reveals important mechanisms of action of Methylphenidate in healthy adults and ADHD children.

I have only a few comments:

- 1) It should be made clearer that the task seems to be also measuring working memory
- 2) It would be important if the authors could add more to the discussion on what the findings actually mean clinically or cognitively. It would be good to add a short discussion of how the network findings might relate to activation effects of Methylphenidate? For example, how do the effects on IFC and basal ganglia relate to findings of activation increases of rIFC and basal ganglia in a meta-analysis of MP effects on fMRI studies in ADHD? (Rubia et al., BPS 2014). How do they relate to FC findings of stronger anticorrelation between the DMN and task-relevant regions?
- 3) Were the subjects blinded to the drug conditions and was the Discovery study also counterbalanced?

Katya Rubia

Reviewer #2

(Remarks to the Author)

The authors demonstrate methylphenidate (MP) compresses the principal unimodal-transmodal gradient compared to placebo (PL). Additional analyses examined gradient compression in relation to PET imaging measures of D1 and D2 receptor availability in these same subjects. They demonstrate the robustness of their findings in a second small replication sample and in adolescents take stimulants in ABCD.

This study has many strengths including an unusually thorough imaging protocol that includes multiple resting state runs under MP and PL, PET imaging in these same subjects, and two additional samples for assessing robustness. An additional strength is the remarkably large effect size for MP effects on regions of the principal gradient such as somatomotor network (Cohen's d 's of around 1). The very large effects sizes provide confidence that they have identified a real, meaningful effect.

Some Comments:

-The head motion protocol is described for the Discovery sample and appears reasonably thorough, and it includes. The authors report no difference in head motion in MP and PL for this sample. The head motion protocol for the Replication sample is not described in detail. Was there a difference in MP and PL in this sample? The same applies to the ABCD sample: head motion approach is not described in detail, and it is unclear if there are head motion differences in kids on stimulants compared to the control sample.

Pg. 6 says "MP administration shifted the principal gradient toward transmodal regions while preserving the overall hierarchical organization of cortical systems observed during PL (Figs. 2a and 2b). Specifically, MP sharpened the principal gradient, further lowering values in the somatomotor cortex and elevating them in the DMN."

I was confused by this claim. Figure 3 a and b with the dense connectomes shows MP increases values in SMN. In parcellated connectomes, we appear to see the same pattern since the text says MP increases values in SMN by Cohen's d of 1. More broadly, it seems correct to say MP compresses the principal gradient and it reduces differentiation of higher-order regions and somatosensory regions. I am unsure of the claim that MP "shifts the gradient towards multimodal areas" (1st paragraph page 8). Multimodal areas show lower values in the principal gradient, so saying the gradient is shifted towards them seems incorrect. Please clarify and perhaps rephrase how the findings are described.

Also, a figure like the first plot in Figure 2d in the Supplement seems like it should be in the main manuscript. That figure conveys a lot of information by showing the expression of the 1st gradient parcel-wide in PL and MPD. I suggest moving that figure to the main manuscript with three changes: 1. Make the x and y axes consistent in their range. 2. Show the $x=y$ line so that it is easier to appreciate what the gradient looks like under MPD relative to the $x=y$ line which is no effect. 3. Color each parcel to reflect its membership in a network, so it is easy to see, for example, that somatomotor parcels and DMN parcels are affected in opposite ways.

Pg. 24 - The authors chose to derive principal gradients from the group average map as well as each subject. An alternative approach is to calculate the principal gradients in another very large dataset such as HCP. Could the author comment on why they chose to use their smaller datasets for deriving principal gradients?

Reviewer #3

(Remarks to the Author)

This is an interesting and rigorous study examining the effects of methylphenidate (MP) on brain function dynamics across multiple datasets. The authors use a diffusion embedding method to examine principal gradients of macroscale cortical organization from resting-state functional connectivity data. The authors found that, across three datasets, the brain organization followed these gradients reliably, and that MP modulated the first principal gradient. They further report data suggesting that these gradients are correlated with MP-induced changes in cognition, and with dopamine receptor availability and norepinephrine transporter distribution. The study is quite complicated, with many different pieces integrated with various degrees of success. Overall, I am generally convinced by the replicability of these gradients and that MP can modulate the patterns. I am less convinced about the functional significance of these effects, as detailed below.

1. It seems intuitively strange that MP would increase gradient values in the DMN. I realize this isn't the same as activation mapping values, but still it would seem that, for MP to enhance attention and cognition in a therapeutic manner, it should shift away from any enhancement of the DMN (task-negative network) and toward the task-positive network. I apologize if I am not understanding this correctly in the context of these analyses, because I am not an expert on them, but I believe I will not be the only reader to wonder about this. Please comment.

2. I am not convinced by the cognition data, where the (uncorrected) correlation appears to be driven by one or more outliers, especially the one at the top right quadrant.

3. I am also left a bit unclear about the functional significance of the correlations with baseline dopamine receptor availability. As the correlation was with a baseline measure, rather than with dopamine transmission per se, I don't know how to think about any MP-induced effects on these data. In this vein, I don't understand the authors' conclusion that their data "supports the idea that functional gradients are not static but can be compressed by neuromodulatory input, particularly dopaminergic stimulation."

4. The examination of NET comes as a surprise in the middle of the results. The Introduction does not properly consider norepinephrine effects.

5. The ABCD data is very unbalanced, with a sample of ~12:1 unmedicated vs. medicated participants. The medicated are also disproportionately boys. Did the authors consider propensity scores as an approach? Furthermore, in the Discussion, the authors do not adequately grapple with the fact that the effect sizes were attenuated in the kids. Is this a developmental effect or a reflection of the larger sample sizes (thus a much smaller "true" effect)? Also, why would it be the case that kids who really need MP – those with ADHD – actually show a watered down effect compared with healthy adults? Shouldn't it be the opposite, in that the kids with ADHD could ostensibly be expected to have a more abnormal baseline gradient that has more room for remediation with MP?

6. It is surprising that it only came up in Methods that a portion of the replication sample had IV MP. This important methodological detail needs to be better integrated into the Results, which come first in this journal format. Does the IV administration have any bearing on the gradients? Please clarify and discuss.

7. Throughout the Figures, the cyan color, which appears quite a bit, is not well-represented in the bars showing the magnitude of the raw effects; rather, these bars tend to show dark blue and green hues only. Please correct this.

Version 1:

Reviewer comments:

Reviewer #1

(Remarks to the Author)

The authors have addressed all comments satisfactorily. This paper will be an important addition to the field.

Reviewer #2

(Remarks to the Author)

The authors have been very responsive. I have no further comments.

Reviewer #3

(Remarks to the Author)

The authors have responded well to reviewer comments.

Reviewer #1:

Overall Comment: "This is a well conducted combined fMRI/PET study that reveals important mechanisms of action of Methylphenidate in healthy adults and ADHD children."

Response: We thank the reviewer for the positive feedback and for recognizing the significance of our combined fMRI/PET approach in elucidating the mechanisms of methylphenidate across both healthy adults and ADHD populations.

Comment 1: "It should be made clearer that the task seems to be also measuring working memory"

Response: We agree and have revised the manuscript to explicitly note that the ball-tracking visual attention task also engages working memory processes (lines 254-255).

Comment 2: "It would be important if the authors could add more to the discussion on what the findings actually mean clinically or cognitively. It would be good to add a short discussion of how the network findings might relate to activation effects of Methylphenidate? For example, how do the effects on IFC and basal ganglia relate to findings of activation increases of rIFC and basal ganglia in a meta-analysis of MP effects on fMRI studies in ADHD? (Rubia et al., BPS 2014). How do they relate to FC findings of stronger anticorrelation between the DMN and task-relevant regions?"

Response: We agree with the reviewer and have added a paragraph in the Discussion to address the potential clinical and cognitive implications of our findings and to relate them to prior activation and functional connectivity studies of methylphenidate. Specifically, we now discuss how the observed MP-induced principal gradient shifts involving the inferior frontal cortex (IFC) and basal ganglia may align with prior reports of increased activation in these regions during cognitive control tasks in ADHD (Rubia et al., 2014), and that though MP's compression of the principal gradient, indicative of reduced cortical segregation across the unimodal transmodal distinction, this does not preclude strengthening of the negative correlation between default-mode and task-positive networks, both of which are networks comprised mostly of transmodal cortices.

Comment 3: "Were the subjects blinded to the drug conditions and was the Discovery study also counterbalanced?"

Response: Yes, all subjects were blinded to the drug conditions, and the Discovery study was fully counterbalanced across placebo and methylphenidate sessions to control for order effects. We have made this clear in the Methods and Results sections (lines 92,95, 240-242, 526, and 537).

Reviewer #2:

Overall Comment: "The authors demonstrate methylphenidate (MP) compresses the principal unimodal-transmodal gradient compared to placebo (PL). Additional analyses examined gradient compression in relation to PET imaging measures of D1 and D2 receptor availability in these same subjects. They demonstrate the robustness of their findings in a second small replication sample and in adolescents take stimulants in ABCD.

This study has many strengths including an unusually thorough imaging protocol that includes multiple resting state runs under MP and PL, PET imaging in these same subjects, and two additional samples for assessing robustness. An additional strength is the remarkably large effect size for MP effects on regions of the principal gradient such as somatomotor network (Cohen's d 's of around 1). The very large effect sizes provide confidence that they have identified a real, meaningful effect."

Response: We sincerely thank the reviewer for the thoughtful and encouraging comments. We greatly appreciate the recognition of the study's strengths, including the comprehensive imaging protocol, multimodal PET/fMRI measures, the use of replication samples, and the large effect sizes observed for MP effects on the principal gradient. We are pleased that our findings are viewed as robust and meaningful, and we hope they contribute to a better understanding of methylphenidate's effects on brain functional organization.

Comment 1: "The authors report no difference in head motion in MP and PL for this sample. The head motion protocol for the Replication sample is not described in detail. Was there a difference in MP and PL in this sample? The same applies to the ABCD sample: head motion approach is not described in detail, and it is unclear if there are head motion differences in kids on stimulants compared to the control sample."

Response: We thank the reviewer for raising this important point. For the Replication cohort, the mean framewise displacement (FD) did not demonstrate significant differences between MP and PL sessions (paired t-test, $P = 0.5$). For the significantly larger sample of the ABCD-BIDS Community Collection 3165, mean FD was slightly higher for stimulant-treated than untreated adolescents (2-sided t-test, $P=0.012$). We have added these methodological details and results to the Results section for clarity (lines 242-243, and 361-363).

Comment 2: "Pg. 6 says 'MP administration shifted the principal gradient toward transmodal regions while preserving the overall hierarchical organization of cortical systems observed during PL (Figs. 2a and 2b). Specifically, MP sharpened the principal gradient, further lowering values in the somatomotor cortex and elevating them in the DMN.' I was confused by this claim. Figure 3 a and b with the dense connectomes shows MP increases values in SMN. In parcellated connectomes, we appear to see the same pattern since the text says MP increases values in SMN by Cohen's d of 1. More broadly, it seems correct to say MP compresses the principal gradient and it reduces differentiation of higher-order regions and somatosensory regions. I am unsure of the claim that MP "shifts the gradient towards multimodal areas" (1st paragraph page 8). Multimodal areas show lower values in the principal gradient, so saying the gradient is shifted towards them seems incorrect. Please clarify and perhaps rephrase how the findings are described."

Response: We thank the reviewer for noting this point and agree our wording was unclear. MP compresses the principal gradient, reducing differentiation between unimodal and transmodal systems by elevating values in SMN and lowering them in DMN, with the reduction in segregation more pronounced in unimodal regions. The term "shift toward multimodal areas" was intended to capture this asymmetry, but we have revised the text to make this meaning explicit (lines 149-157, 180, 190, and 256).

Comment 3: “Also, a figure like the first plot in Figure 2d in the Supplement seems like it should be in the main manuscript. That figure conveys a lot of information by showing the expression of the 1st gradient parcel-wide in PL and MPD. I suggest moving that figure to the main manuscript with three changes: 1. Make the x and y axes consistent in their range. 2. Show the x=y line so that it is easier to appreciate what the gradient looks like under MPD relative to the x=y line which is no effect. 3. Color each parcel to reflect its membership in a network, so it is easy to see, for example, that somatomotor parcels and DMN parcels are affected in opposite ways.

Response: We thank the reviewer for the suggestion. We added a new Figure 4 to the main manuscript showing the parcel-wise scatterplot of the first gradient, with plots scaled and data color-coded by network membership as suggested. In the Results, we now report that attenuation of somatomotor (SMM) network segregation in the principal gradient was more pronounced than in other networks, whereas no significant changes were observed in the secondary gradient (lines 198-205, and 225-228).

Comment 4: “Pg. 24 - The authors chose to derive principal gradients from the group average map as well as each subject. An alternative approach is to calculate the principal gradients in another very large dataset such as HCP. Could the author comment on why they chose to use their smaller datasets for deriving principal gradients?”

Response: We chose to derive principal gradients from our own group-average and individual connectivity matrices to ensure that the gradient axes reflected the connectivity structure of our specific cohorts and acquisition protocols, thereby maximizing sensitivity to MP-related changes. While deriving gradients from a large external dataset such as HCP could provide highly stable axes, differences in acquisition parameters, preprocessing, and participant characteristics may introduce alignment challenges and reduce sensitivity to within-study effects. Nonetheless, our results are broadly consistent with gradients reported in large normative datasets, supporting the generalizability of our findings.

Reviewer #3:

Overall comment: “This is an interesting and rigorous study examining the effects of methylphenidate (MP) on brain function dynamics across multiple datasets. The authors use a diffusion embedding method to examine principal gradients of macroscale cortical organization from resting-state functional connectivity data. The authors found that, across three datasets, the brain organization followed these gradients reliably, and that MP modulated the first principal gradient. They further report data suggesting that these gradients are correlated with MP-induced changes in cognition, and with dopamine receptor availability and norepinephrine transporter distribution. The study is quite complicated, with many different pieces integrated with various degrees of success. Overall, I am generally convinced by the replicability of these gradients and that MP can modulate the patterns.”

Response: We thank the reviewer for their thoughtful and constructive comments. We appreciate their recognition of the rigor of our multimodal approach and the replicability

of our findings across datasets. Our goal was precisely to integrate multiple levels of analysis (functional gradients, cognition, and molecular imaging) to provide converging evidence for MP's effects on large-scale brain organization. We are encouraged that the reviewer is generally convinced by both the robustness of the gradients and the modulation of these patterns by MP.

Comment 1: "It seems intuitively strange that MP would increase gradient values in the DMN. I realize this isn't the same as activation mapping values, but still it would seem that, for MP to enhance attention and cognition in a therapeutic manner, it should shift away from any enhancement of the DMN (task-negative network) and toward the task-positive network. I apologize if I am not understanding this correctly in the context of these analyses, because I am not an expert on them, but I believe I will not be the only reader to wonder about this. Please comment."

Response: Our findings do not indicate that MP "enhances" DMN function; rather, MP compressed the principal gradient, reducing the segregation between unimodal and transmodal systems, including the DMN. This reduction in segregation reflects greater integration across cortical systems, which we interpret as a mechanism supporting MP's effects on attention and cognition (lines 224-225).

Comment 2: "I am not convinced by the cognition data, where the (uncorrected) correlation appears to be driven by one or more outliers, especially the one at the top right quadrant."

Response: We believe that, despite the lack of corrections for multiple comparisons in the correlations between MP-induced changes in VA accuracy and changes in the strength of the principal gradient (Figs. 5c and 5d), these results are important for interpreting the main findings. Accordingly, we now present them as exploratory in the main text, while noting their interpretive limitations (lines 252, 258, and 421), and the corresponding Figure was moved to the supplement.

Comment 3. "I am also left a bit unclear about the functional significance of the correlations with baseline dopamine receptor availability. As the correlation was with a baseline measure, rather than with dopamine transmission per se, I don't know how to think about any MP-induced effects on these data. In this vein, I don't understand the authors' conclusion that their data 'supports the idea that functional gradients are not static but can be compressed by neuromodulatory input, particularly dopaminergic stimulation.'"

Response: We appreciate the reviewer's concern. While our correlations were with baseline D1R/D2R availability rather than dopamine release directly, we view these findings as functionally meaningful because baseline receptor architecture reflects the dopaminergic system's organization, which constrains how cortical gradients respond to MP. MP acts primarily on DAT and NET, increasing extracellular dopamine and norepinephrine, and the baseline D1R/D2R measures likely shape interindividual differences in how the brain reorganizes under this pharmacological challenge. Baseline receptor measures are also typically more stable and have a higher signal-to-noise ratio than differential release measures, which may account for the stronger and more consistent associations. Our conclusion that functional gradients can be compressed by

neuromodulatory input is based primarily on the MP vs. PL comparisons, which revealed robust state-dependent reorganization of the principal gradient. The correlations with baseline D1R/D2R availability are complementary: they highlight how trait-like dopaminergic architecture influences the degree of MP-induced reorganization, but they do not by themselves imply neuromodulatory effects. The revised discussion addresses these issues (lines 442, and 449-454).

Comment 4. “The examination of NET comes as a surprise in the middle of the results. The Introduction does not properly consider norepinephrine effects.”

Response: We have revised the Introduction to clarify that methylphenidate (MP) blocks both DAT and NET, leading to increases in both dopamine and norepinephrine, providing the rationale for examining NET alongside dopaminergic measures (lines 39-42). To emphasize these findings, Figure 7 now highlights the associations with NET density in the main text (lines 347-454).

Comment 5. “The ABCD data is very unbalanced, with a sample of ~12:1 unmedicated vs. medicated participants. The medicated are also disproportionately boys. Did the authors consider propensity scores as an approach? Furthermore, in the Discussion, the authors do not adequately grapple with the fact that the effect sizes were attenuated in the kids. Is this a developmental effect or a reflection of the larger sample sizes (thus a much smaller “true” effect)? Also, why would it be the case that kids who really need MP – those with ADHD – actually show a watered down effect compared with healthy adults? Shouldn’t it be the opposite, in that the kids with ADHD could ostensibly be expected to have a more abnormal baseline gradient that has more room for remediation with MP?”

Response: To address the imbalance in the ABCD sample, we conducted a factorial ANCOVA, including inattention, age, and head motion as continuous covariates, and sex, research site, and scanner type as categorical covariates. This approach allowed us to account for potential confounding factors without relying on propensity scores. Our focus was on replicating the main patterns observed in our discovery sample rather than estimating causal effects of medication status, and the results remained robust when controlling for these covariates. The attenuated effect sizes in the ABCD children likely reflect multiple factors. First, developmental differences in functional connectome organization may limit the magnitude of MP-induced gradient shifts relative to adults, as brain networks are still maturing. Second, the ABCD stimulant-medicated sample includes children taking MP as well as other ADHD medications, introducing heterogeneity that likely reduces observable effect sizes. Third, the ABCD dataset is cross-sectional rather than within-subject, limiting sensitivity to drug-induced changes. Finally, while children with ADHD might be expected to show more abnormal baseline gradients, variability in symptom severity, comorbidities, doses and type of medication likely dilutes the observable group-level effects. These factors together likely account for the smaller effects in children without contradicting the mechanistic role of MP. The revised Discussion addresses these issues (lines 490-492).

Comment 6. “It is surprising that it only came up in Methods that a portion of the replication sample had IV MP. This important methodological detail needs to be better

integrated into the Results, which come first in this journal format. Does the IV administration have any bearing on the gradients? Please clarify and discuss.”

Response: We have clarified in the Methods that a portion of the replication sample received IV MP. While preliminary checks indicate that IV MP produces effects on the principal gradient consistent with oral MP, we do not report these data in the current manuscript, as IV MP is not clinically used for ADHD. This has been clarified in Methods (lines 538-539). This approach ensures transparency regarding the sample while maintaining the clinical and conceptual focus. We aim to fully report IV-MP effects separately in a future study.

Comment 7. “Throughout the Figures, the cyan color, which appears quite a bit, is not well-represented in the bars showing the magnitude of the raw effects; rather, these bars tend to show dark blue and green hues only. Please correct this.”

Response: We thank the reviewer for pointing this out. We have adjusted the color palette in the revised figures (Figures 2,3, and 5) to ensure that cyan is clearly distinguishable and consistently represented in both the maps and the bars.